plant science/molecular biology

ethylene, *CoACO* genes, fruit abscission, *Camellia oleifera*

# Ethylene-regulated immature fruit abscission is associated with higher expression of *CoACO* genes in *Camellia oleifera*

Xiao Hu[1,†], Mi Yang[1,†], Shoufu Gong[2], Hongbo Li[1], Jian Zhang[2], Muhammad Sajjad[3], Xiaoling Ma[1] and Deyi Yuan[1]

[1]Key Laboratory of Cultivation and Protection for Non-Wood Forest Trees of Ministry of Education and the Key Laboratory of Non-Wood Forest Products of Forestry Ministry, Central South University of Forestry and Technology, Changsha 410004, People's Republic of China
[2]Xinyang Agriculture and Forestry University, Xinyang 464000, People's Republic of China
[3]Department of Biosciences, COMSATS University Islamabad (CUI), Park Road, Islamabad 45550, Pakistan

XM, 0000-0002-1741-8978

**Authors for correspondence:**
Xiaoling Ma
e-mail: fanxiaoling@163.com
Deyi Yuan
e-mail: yuan-deyi@163.com

[†]These authors contributed equally to this study.

Immature fruit abscission is a key limiting factor in *Camellia oleifera* Abel. (*C. oleifera*) yield. Ethylene is considered to be an important phytohormone in regulating fruit abscission. However, the molecular mechanism of ethylene in regulating fruit abscission in *C. oleifera* has not yet been studied. Here, we found that the 1-aminocyclopropane-1-carboxylic acid (ACC) content was significantly increased in the abscission zones (AZs) of abnormal fruits (AF) which were about to abscise when compared with normal fruits (NF) in *C. oleifera* 'Huashuo'. Furthermore, exogenous ethephon treatment stimulated fruit abscission. The cumulative rates of fruit abscission in ethephon-treated fruits (ETH-F) on the 4th (35.0%), 8th (48.7%) and 16th (57.7%) days after treatment (DAT) were significantly higher than the control. The ACC content and 1-aminocyclopropane-1-carboxylate oxidase (ACO) activity in AZs of ETH-F were also significantly increased when compared with NF on the 4th and 8th DAT. *CoACO1* and *CoACO2* were isolated in *C. oleifera* for the first time. The expressions of *CoACO1* and *CoACO2* were considerably upregulated in AZs of AF and ETH-F. This study suggested that ethylene played an important role in immature fruit abscission of *C. oleifera* and the two *CoACOs* were the critical genes involved in ethylene's regulatory role.

# 1. Introduction

*Camellia oleifera* Abel. (*C. oleifera*) is widely cultivated in southern China. It is one of the top four (oil palm, olive, coconut, camellia) globally grown woody oil plants [1]. *Camellia oleifera* oil is rich in unsaturated fatty acid (approx. 75–83% oleic acid and 7–13% linoleic acid) [2], and these acids can reduce the risk of cardiovascular diseases [3–5]. In China, the cultivation area of *C. oleifera* is expanding with fruit abscission as the major yield-limiting factor.

Fruit abscission is a developmentally regulated and complicated process, occurring in the abscission zone (AZ) generally located at the base of the pedicel of fruits during fruit ripening or environmental stresses [6–8]. The three obvious abscission stages in *C. oleifera* are flowering stage, August stage and preharvest stage. The flower and fruit abscission rates are very high, ranging between 10–25% and 15–20%, respectively, which seriously affect the yield in *C. oleifera*. Phytohormones are generally considered to be involved in fruit abscission, and ethylene is one of the key hormones [9]. Ethephon (2-chloroethylphosphate acid) is an ethylene-releasing molecule which is stable in a low pH solution, but it hydrolyses in the higher pH of plant tissues releasing ethylene [10]. Ethylene is a gaseous plant growth regulator that can promote fruit ripening, affect fruit colour, firmness, carbon dioxide production and other physiological properties [11]. Exogenously applied ethephon stimulates ethylene production and triggers ethylene-dependent reactions such as flower or fruit abscission [12]. It has been shown that ethylene is the main signal molecule to induce fruit abscission in apple, oil palm and litchi [13–15]. Moreover, studies have shown that the fruit-setting rates can be improved by applying ethylene inhibitor AVG in apple [16]. In general, ethylene biosynthesis may be involved in the abscission process, and it plays an important role in the final stage of abscission. In the process of the biosynthesis of ethylene, methionine is converted to *S*-adenosyl-methionine (SAM). The conversions of SAM to 1-aminocyclopropane-1-carboxylic acid (ACC) and ACC to ethylene are catalysed by ACC synthase (ACS) and ACC oxidase (ACO), respectively [17]. The importance of ACO in regulating the production of ethylene in plants has been proved gradually. ACO is also the rate-limiting step in the biosynthesis of ethylene as ACS. ACO plays an important role in the post-climacteric ripening of tomato fruits. ACO is encoded by multigene families [18]. For example, many *ACO* genes were identified in apple (7), *Arabidopsis* (5), tomato (7), rice (9) and maize (13) [19].

The ethylene biosynthesis could effectively be regulated by controlling the expression level of *ACO* genes [17]. *OeACO2* was upregulated during mature fruit abscission in olive [20]. The expression of *MdACO1* and *MdACO2* at a high level was reported in peel from fruitlets undergoing abscission in apple [21]. The research on preharvest fruit abscission in apple indicated that the pattern of *MdACO1* showed differences between 'Golden Delicious' (susceptible to preharvest fruit abscission) and 'Fuji' (no preharvest fruit abscission). *MdACO1* was upregulated in fruit peel and AZ of 'Golden Delicious', whereas the increase in expression level of *MdACO1* was detected only in fruit peel of 'Fuji' [22]. Furthermore, previous studies showed that ethephon induced the expression of *ACO* [23,24]. Meanwhile, the expression of *ACO* genes was regulated by ethylene such as three *AtACO* genes in *Arabidopsis*.

There are few reports on the mechanism of immature fruit abscission in *C. oleifera*. Here, the internal structure of AZs of NF, AF and ethephon-treated fruits (ETH-F) was observed by scanning electron microscopy. The ACC content and ACO activity in AZs of the three types of fruits were studied. The objective of the present study was to dissect the relationship between ethylene and fruit abscission in *C. oleifera*. We also isolated the *CoACO* cDNAs and examined their expression levels in different AZs. The identification of *CoACO* genes underpinning fruit abscission in *C. oleifera* during the August stage and the ethephon-induced fruit abscission mechanism will provide valuable information for fruit abscission studies in *C. oleifera*.

# 2. Material and methods

## 2.1. Plant materials and treatments

Uniform 9-year-old, high-crown grafted *C. oleifera* cultivar 'Huashuo' trees (growing in an orchard located at Wangcheng, Hunan province, China, under local cultivation and management conditions) were used in this study. Young roots, young stems, young leaves, young fruits, fruit pedicels and fruit AZs (FAZs) were sampled on 16 April 2019. Each sample had three biological replicates.

To investigate the physiological and molecular mechanism of fruit abscission in *C. oleifera*, AZs of NF and AF from 9-year-old 'Huashuo' trees were collected for gene expression and ACC content and ACO

activity measurements, during the critical August abscission stage. Besides, the fruit abscission situation was observed in ethephon treatment. Three 'Huashuo' trees were randomly selected for each of the following treatments: water (control) and $2 \, g \, l^{-1}$ ethephon solution (earlier experimental results showed that $2 \, g \, l^{-1}$ ethephon could significantly increase the fruit abscission rates when compared with the control and the trees were not damaged). All treatments were sprayed to run-off with 0.05% Tween 20 solution using a motorized back-pack sprayer on 22 August. For each experimental tree, four branches (about 30 fruits in each branch) were tagged to count fruit abscission rates on the 4th, 8th and 16th days after treatment (DAT), whereas the rest of the branches were used for taking FAZ samples for gene expression and ACC content and ACO activity measurements. FAZs were collected by cutting 1 mm at each side of the abscission fracture plane. Fifteen FAZs of each type were sampled per tree. Each tree was treated as a biological replicate. Each sample had three biological replicates. All samples were immediately snap frozen in liquid nitrogen and stored at $-80°C$ until further processing.

## 2.2. Scanning electron microscopy

The AZs (fracture surface belongs to the branch) of the NF, AF and ETH-F (the 8th DAT) were collected. All samples used in SEM were collected on the same day. So, the AZs of NF can be used as the control of ETH-F. We can compare the differences among these three types of samples. These samples were fixed for 3 h in 2.5% (v/v) glutaraldehyde in $0.1 \, mol \, l^{-1}$ sodium phosphate buffer (pH = 7.2). Following thorough washing in $0.1 \, mol \, l^{-1}$ sodium phosphate buffer, samples were post-fixed in 1% (w/v) osmium tetroxide for 2 h and rinsed with the same buffer. Dehydration was carried out in a graded series of ethanol. For SEM, 100% ethanol was replaced with 3-methylbutyl acetate. Samples were critical-point dried, sputter-coated with platinum and observed the fracture plane of different samples under a scanning electron microscope (Zeiss Supra 10 vp; Carl Zeiss Microscope, NY, USA).

## 2.3. ACC content and ACO activity measurements

For ACC content and ACO activity measurements, 1 g FAZs from each biological replicate was homogenized in phosphate-buffered saline, and the homogenate was centrifuged at $12\,000g$ for 20 min at 4°C. The supernatant was used for enzyme assay according to procedures described in the instructions of Plant 1-aminocyclopropane carboxylic acid ACC Kit and Plant ethylene oxidase (ACO) enzyme linked immunosorbent assay kit (Jiangxi gelatins biological reagent company). Every measurement had three biological replicates.

## 2.4. RNA extraction and cDNA synthesis

The FAZs of *C. oleifera* were manually homogenized and total RNA was isolated according to procedures described in the instructions of HP Plant RNA Kit (Omega Bio-Tek). First-strand cDNA was synthesized from 1 µg of total RNA using the HiScript II Q RT SuperMix (Vazyme Biotech Co., Ltd) following the manufacturer's instructions.

## 2.5. Isolation of *CoACO* genes

The coding sequences of the tea plant tree *CsACO* genes were used as a query for BLAST searchers against the transcriptome data of FAZ in *C. oleifera* (X. Ma *et al.*, 2021, unpublished data) to acquire the *CoACO* cDNA sequences. All obtained sequences with high similarity to the *CsACO* genes constructed putative *CoACO* genes using DNAMAN (http://www.lynnon.com). Based on the sequences, the specific primer pairs (table 1) were designed to amplify *CoACO* genes. Each PCR reaction mixture, containing 1 µl of obtained cDNA, 1 µl of primers solution (10 µM), 10 µl of $2 \times Es$ Taq MasterMix (Dye) and 8 µl RNase-free water to a total volume of 20 µl, was subjected to the thermal cycling conditions as follows: 3 min at 95°C; 35 cycles of 30 s at 95°C, 30 s at 55°C and 60 s at 72°C; and a final extension of 5 min at 72°C. The PCR products (20 µl) were separated in 1% agarose gel, and were sequenced by T$_{SING}$K$_E$ Biological Technology company.

## 2.6. Sequence analysis

Sequences were retrieved using the GeneBank Blast (http://www.ncbi.nlm.nih.gov/blast/). Sequences were analysed by MegAlign software. Physical and chemical parameters of proteins were determined

**Table 1.** Primer sequences of the genes used in this research.

| primer name | primer sequence (5′–3′) | annealing temperature (°C) | PCR product size | function |
|---|---|---|---|---|
| CoACO1-F | TGTGAGAACTGGGGCTTTTT | 59.7 | 1057 bp | gene cloning |
| CoACO1-R | ACCACAACCAAACACCCAAT | 60 | | |
| CoACO2-F | ATGGAGGCCTTCCCCACTT | 59.5 | 966 bp | gene cloning |
| CoACO2-R | TTAATAAGCTGTTGCAACAGTG | 55.9 | | |
| eCoACO1-F | TTCCTCCGATGAAACACTCC | 60.1 | 173 bp | expression analysis |
| eCoACO1-R | AGGTGCCGGATAGATGACAG | 60.1 | | |
| eCoACO2-F | CACCAATGGCAAATACAAG | 51.9 | 207 bp | expression analysis |
| eCoACO2-R | TTCAACCCAGCATAGAGC | 51 | | |
| EF1α-F | AGACTGTGGCTGTTGGTGTT | 58 | about 200 bp | internal control |
| EF1α-R | ATCCAAACCCGCACAGTTCA | 60 | | |

using ProtParam tool (http://us.expasy.Org/tools/protparam.html). The predicted CoACO1 and CoACO2 proteins were BLASTed in the NCBI CDD system (https://www.ncbi.nlm.nih.gov/Structure/cdd/wrpsb.cgi) to search for conserved domains. Multiple sequence alignment of CoACO1 and CoACO2 with grape ACO proteins was carried out using MUSCLE with default parameters [25]. We determined the best substitution model of all ACO protein sequences by using ModelFinder [26] and found that the best fit for these proteins was LG + G4. The IQ-tree software was used to perform maximum-likelihood (ML) phylogenetic analysis with a bootstrap test for 1000 replicates and an SH-aLRT test for 1000 random addition replicates [27]. The phylogenetic tree was constructed using the FigTree v. 1.4.3 software.

## 2.7. Quantitative real-time PCR analysis

The *CoACO* genes expression was performed for different tissues, AZs of NF, AF and ETH-F. The quantitative real-time PCR (qRT-PCR) was carried out on a BIO-RAD CFX96™ Real-Time System. Gene-specific primers were designed using Primer 3.0 software listed in table 1. The qRT-PCR was under the following conditions: 95°C for 1 min; 40 cycles at 95°C for 10 s, 55°C for 10 s, 72°C for 15 s. The qRT-PCR mixtures consisted of 2 µl cDNA, 1.6 µl primers solution, 10 µl 2 × $T_{SING}K_E$ Master qPCR Mix and 6.4 µl ddH$_2$O. Gene relative expression levels were calculated by the $2^{-\Delta Ct}$ comparative threshold cycle (Ct) method.

## 2.8. Statistical analysis

Data were analysed by one-way analysis of variance (ANOVA) with three replications of each experiment. The Duncan's multiple range test was applied at $p = 0.05$ level to evaluate the significant differences among the mean values.

# 3. Results

## 3.1. Exogenous ethephon stimulates fruit abscission in *C. oleifera*

We examined the effect of ethephon on the fruit abscission in *C. oleifera*. The ethephon treatment significantly promoted the fruit abscission and its cumulative fruit abscission rates were always significantly higher than that of control (figure 1). On the 4th DAT, the cumulative rates of fruit abscission in ETH-F were 35.0% with about ninefold higher than that of the control and up gradually to 57.7% on the 16th DAT. The rate of fruit abscission in ETH-F reached its maximum on the 4th DAT. However, the cumulative fruit abscission rate in the control increased slowly until the 8th DAT and up rapidly to 25.0% on the 16th DAT (figure 1). The reason is that 'Huashuo' during this period was at the peak of fruit abscission. The fruit abscission at this stage was named as the August stage in *C. oleifera*.

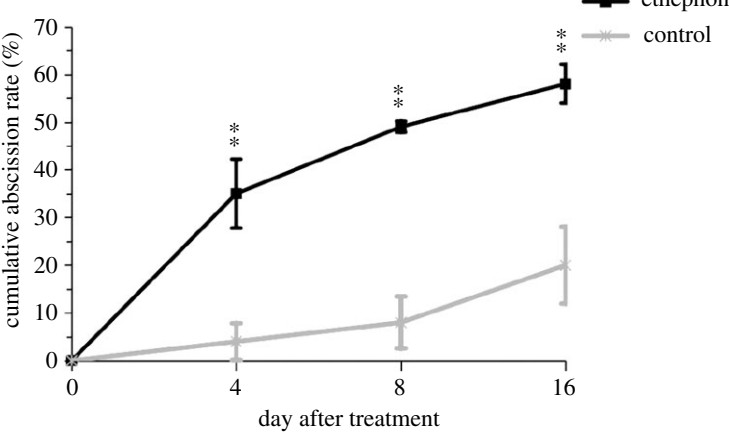

**Figure 1.** Effects of ethephon treatment on fruit abscission of *C. oleifera* 'Huashuo'. Three biological replicates for each treatment were averaged. Error bars indicate the s.d. of the means ($n = 3$). One-way ANOVA (Tukey–Kramer test) analysis was performed, and statistically significant differences ($p < 0.01$) were indicated by double asterisks in the ethephon treatment with respect to the control.

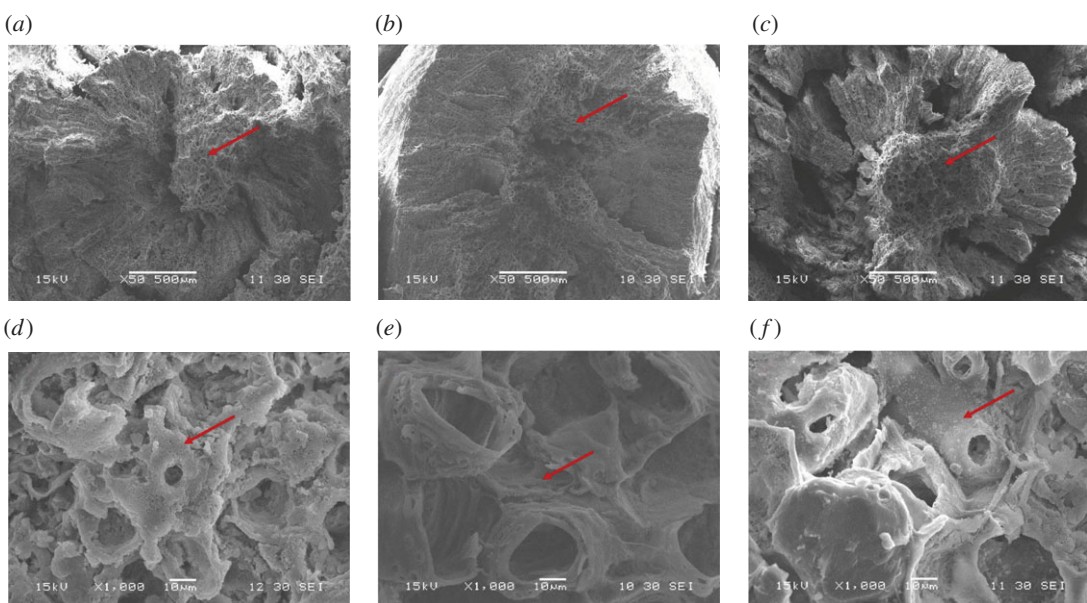

**Figure 2.** Scanning electron micrographs of the stem AZs structures of NF (*a,d*), AF (*b,e*) and ETH-F (*c,f*) of *C. oleifera* 'Huashuo'. (*a,d*) SEM of the stem AZs structure of NF. NF was showed as the natural status. The entire abscission fracture plane was convex and severely tearing. The surface of the pith contained many broken and rough tracheary elements. (*b,e*) SEM of the stem AZs structure of AF. AF was showed as the abscission status. This type of fruit will soon fall off. The abscission fracture plane of AF was divided into two layers: the pith and its surrounding vascular tissue. Only the pith of AF showed the weak adhesion between the fruit stalk and stem. The surface of the pith was smooth, and no inclusions were released. (*c,f*) SEM of the stem AZs structure of ETH-F. ETH-F was shown as the other abscission status. The types of fruits sampled on the 8th DAT of ETH did not yet show the state of AF. The abscission fracture plane of ETH-F was uneven. There was a small amount of material on the surface of the pith. *Bars* in *a–c* represent 500 µm and *bars* in *d–f* represent 10 µm. The arrows indicate the pith of the abscission fracture plane.

## 3.2. The fracture surface of FAZs prior to abscission undergoes structural changes

To analyse the changes of AZs structure of fruits prior to abscission (AF and ETH-F) with respect to NF, SEM was used to observe the abscission fracture plane with one natural status (NF) and two abscission status (AF and ETH-F). The AZ of *C. oleifera* is formed at the junction of fruit stalk and stem. The basic differences in the abscission fracture plane related to the adhesion between fruit stalk and stem among the NF (figure 2*a*) and fruits which were about to abscission (AF and ETH-F) (figure 2*b,c*). The abscission fracture plane of NF, AF and ETH-F was taken off artificially. The entire abscission fracture plane was convex and severely tearing and had a layered appearance, since the adhesion of fruit stalk and stem was tight in the AZs of NF (figure 2*a*). The abscission fracture plane of ETH-F with the middle pith

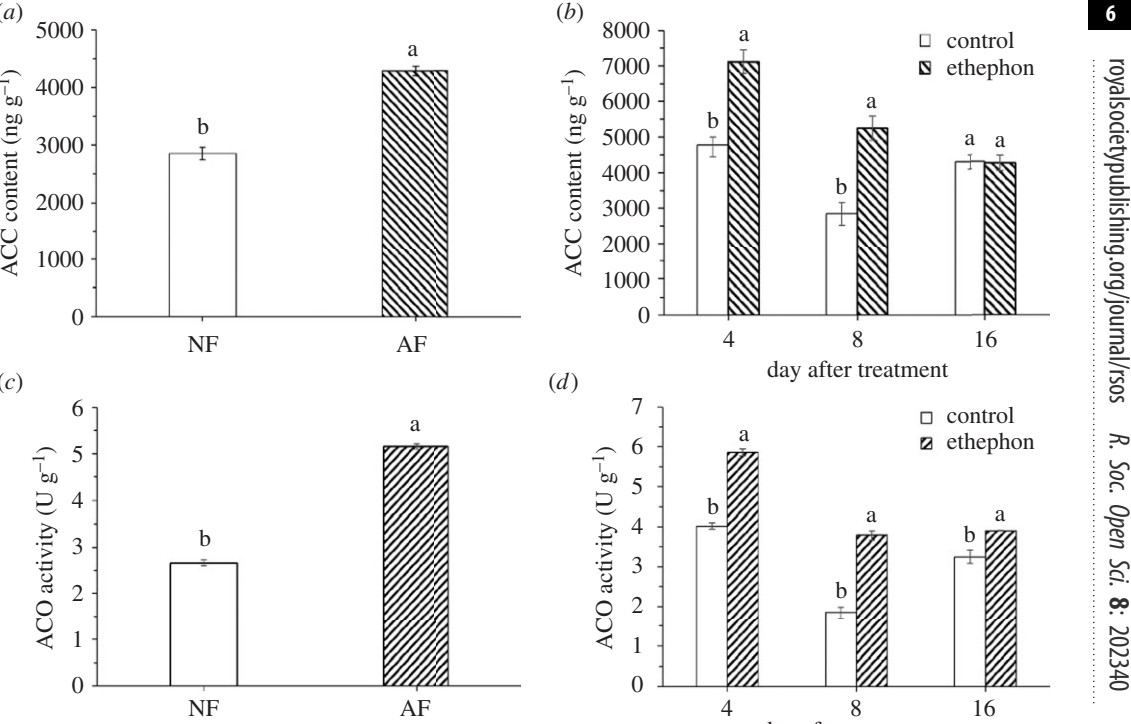

**Figure 3.** ACC content and ACO activity in FAZs of *C. oleifera* 'Huashuo'. (*a*) ACC content in AZs of NF and AF. (*b*) ACC content in FAZs after ethephon treatment. (*c*) ACO activity in AZs of NF and AF. (*d*) ACO activity in FAZs after ethephon treatment. Three biological replicates for each treatment were averaged. Error bars indicate the s.d. of the means (*n* = 3). One-way ANOVA (Duncan's multiple range test) analysis was performed, and statistically significant differences ($p < 0.05$) were indicated by different letters.

and part of its surrounding vascular tissue was uneven and slightly tearing (figure 2*c*). The adhesion between fruit stalk and branch decreased on the 8th DAT of ETH-F with respect to the NF. The abscission fracture plane of AF was divided into two layers: the pith and its surrounding vascular tissue. The vascular tissue between fruit stalk and stem had been isolated. The adhesion of fruit stalk and stem in the pith of AF was the weakest among all three types (figure 2*b*). Therefore, the pith of NF, AF and ETH-F was deeply observed at a magnification of 1000 times. The broken and rough tracheary elements could be clearly observed in the fracture surface of NF (figure 2*d*), while the elements in the fracture surface of AF were smooth (figure 2*e*), and that of ETH-F was somewhere in between (figure 2*f*) indicating that the fruits sampled on the 8th DAT of ETH did not reach the state of AF. In summary, the abscission fracture plane structures of two abscission statuses (AF and ETH-F) have changed accordingly with respect to NF.

## 3.3. ACC content and ACO activity in AZs of NF, AF and ETH-F

To further determine the relationship between ethylene and fruit abscission, we analysed the ACC content and ACO activity in FAZ tissues of two kinds of comparisons, namely NF and AF, ETH-F, and control during fruit abscission of the August stage in *C. oleifera*. The ACC content and ACO activity were significantly increased in AZ tissues of AF to about 1.5-fold and 1.9-fold higher than that of NF, respectively (figure 3*a,c*). Besides, the ACC content in AZs of ETH-F was significantly increased to about 1.5-fold and 1.8-fold higher than that of control on the 4th and 8th DAT, respectively, whereas there was no significant difference on the 16th DAT (figure 3*b*). The ACO activity in AZs of ETH-F was significantly increased to about 1.5-fold, 2.1-fold and 1.2-fold higher than that of control on the 4th, 8th and 16th DAT, respectively (figure 3*d*). AF and ETH-F were two types of fruits which were about to abscission. The ACC content and ACO activity in AZs of AF and ETH-F changed with respect to the NF and control, respectively. These results show that ethylene plays an important role in fruit abscission of the August stage.

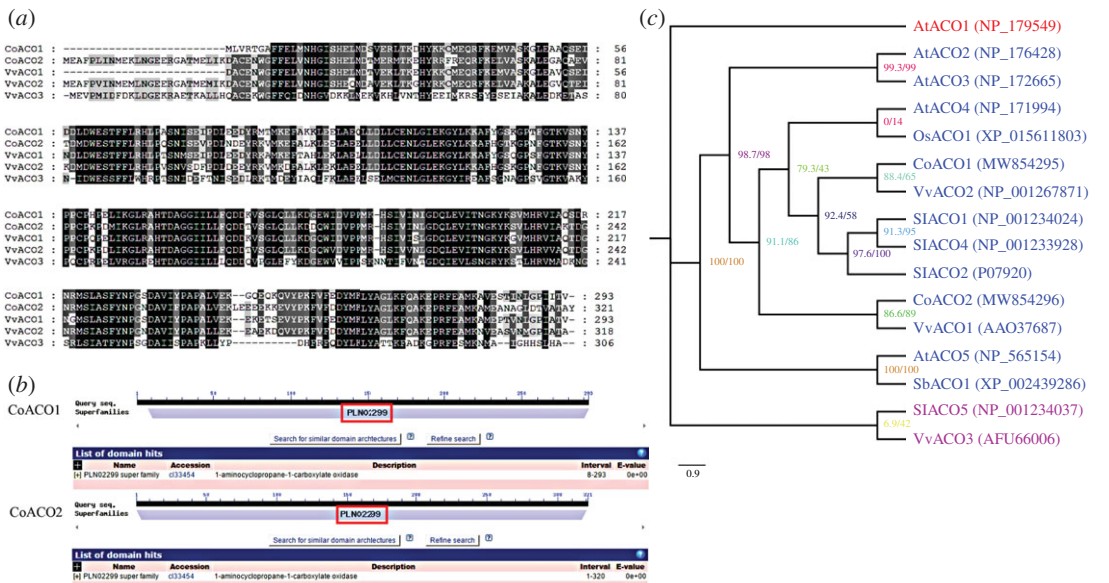

**Figure 4.** Characterization of the CoACO proteins. (*a*) Alignment of the deduced amino acid sequences of CoACO1 and CoACO2 with grape ACO 1–3 proteins. Amino acid position numbers are indicated to the right of the sequence. (*b*) Putative domain analysis of the CoACO1 and CoACO2 proteins. The conserved domain of the PLN0229 super family was marked with red rectangles. (*c*) Phylogenetic analysis of the ACO protein family. A maximum-likelihood phylogenetic tree was constructed using the ACO protein sequences from *Arabidopsis thaliana* (*At*), *Vitis vinifera* (*Vv*), *Solanum lycopersicum* (*Sl*), *sorghum bicolor* (*Sb*) and *Oryza sativa* (*Os*). The 16 proteins were clustered into the three distinct groups indicated. The accession numbers of ACO proteins in GenBank are: CoACO1 (MW854295), CoACO2 (MW854296), AtACO1 (NP_179549), AtACO2 (NP_176428), AtACO3(NP_172665), AtACO4 (NP_171994), AtACO5 (NP_565154), OsACO1 (XP_015611803), VvACO1 (AAO37687), VvACO2 (NP_001267871), VvACO3 (AFU66006), SlACO1 (NP_001234024), SlACO2 (P07920), SlACO4 (NP_001233928), SlACO5 (NP_001234037), SbACO1 (XP_002439286). All amino acid sequences were retrieved from GenBank databases.

## 3.4. Isolation and sequence analysis of *CoACO* cDNAs

Through BLAST analysis based on *CsACO* sequences, we identified cDNA sequences of two *ACO* genes in the *C. oleifera* 'Huoshuo' that were highly similar to *CsACO* genes. Here, we designated them as *CoACO* genes. Based on the PCR and sequencing, two *CoACO* genes from 'Huashuo' were isolated and named as *CoACO1* and *CoACO2*. The CDS of *CoACO1* and *CoACO2* were 882 and 966 bp in length, encoding a deduced protein of 293 and 321 amino acids, respectively. Corresponding molecular weights (MWs) for the deduced CoACO1 and CoACO2 protein were 33.27 and 36.64 kDa, respectively.

ACO protein alignment between oil-tea camellia and grape was conducted based on their full-length protein sequences (figure 4). The amino acid sequence similarity between CoACO1 and CoACO2 was 70.5%. The amino acid sequence similarities among CoACO1 and VvACO1, VvACO2 and VvACO3 were 77.6%, 71.4% and 43.8%, respectively. The amino acid sequence similarities among CoACO2 and VvACO1, VvACO2 and VvACO3 were 73.6%, 84.5% and 46.9%, respectively (figure 4*a*). The result of putative domain analysis showed that both the CoACO1 and CoACO2 proteins included PLN02299 super family conserved regions which characterized ACO in other species (figure 4*b*).

To examine the relationship of oil-tea camellia *ACO* genes with those in other species, a phylogenetic analysis using the full-length ACO protein sequences from *Arabidopsis* (*At*), grape (*Vv*), tomato (*Sl*), sorghum (*Sb*) and rice (*Os*) was performed. As shown in figure 4*c*, the ACO family was divided into three subgroups. CoACO1 and CoACO2 belonged to the same group. CoACO1 showed closer evolutionary relationship with VvACO1, whereas CoACO2 was closer to VvACO2.

## 3.5. Gene expression analysis of *CoACOs* in different tissues and in AZs of NF, AF and ETH-F

The qRT-PCR was conducted to investigate the expression profile of *CoACO* genes in different tissues of *C. oleifera* 'Huashuo' in April. The results showed that *CoACO* genes can express in various young tissues: roots, stems, leaves, fruits, fruit pedicels and FAZs with different expression patterns. Expression of *CoACO1* was the highest in fruits and the lowest in fruit pedicels (figure 5*a*), whereas the highest expression level of *CoACO2* was in fruit pedicels, and the lowest in young stems (figure 5*b*). Besides, both

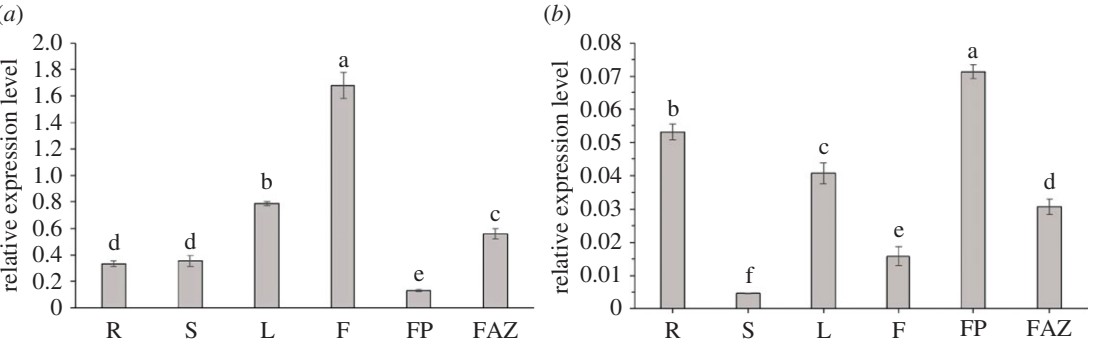

**Figure 5.** Relative expression levels of *CoACO* genes in *C. oleifera* 'Huashuo' tissues by qRT-PCR. (*a,b*) qRT-PCR analysis of *CoACO1* and *CoACO2* expression, respectively, in young fruits (F), roots (R), stems (S), leaves (L), fruit pedicles (FP) and fruit abscission zones (FAZ). *CoEF-1a* was used as an internal control. Three biological replicates for each sample were averaged. Error bars indicate the s.d. of the means ($n = 3$). One-way ANOVA (Tukey *post hoc* test) analysis was performed, and statistically significant differences ($p < 0.05$) were indicated by different letters.

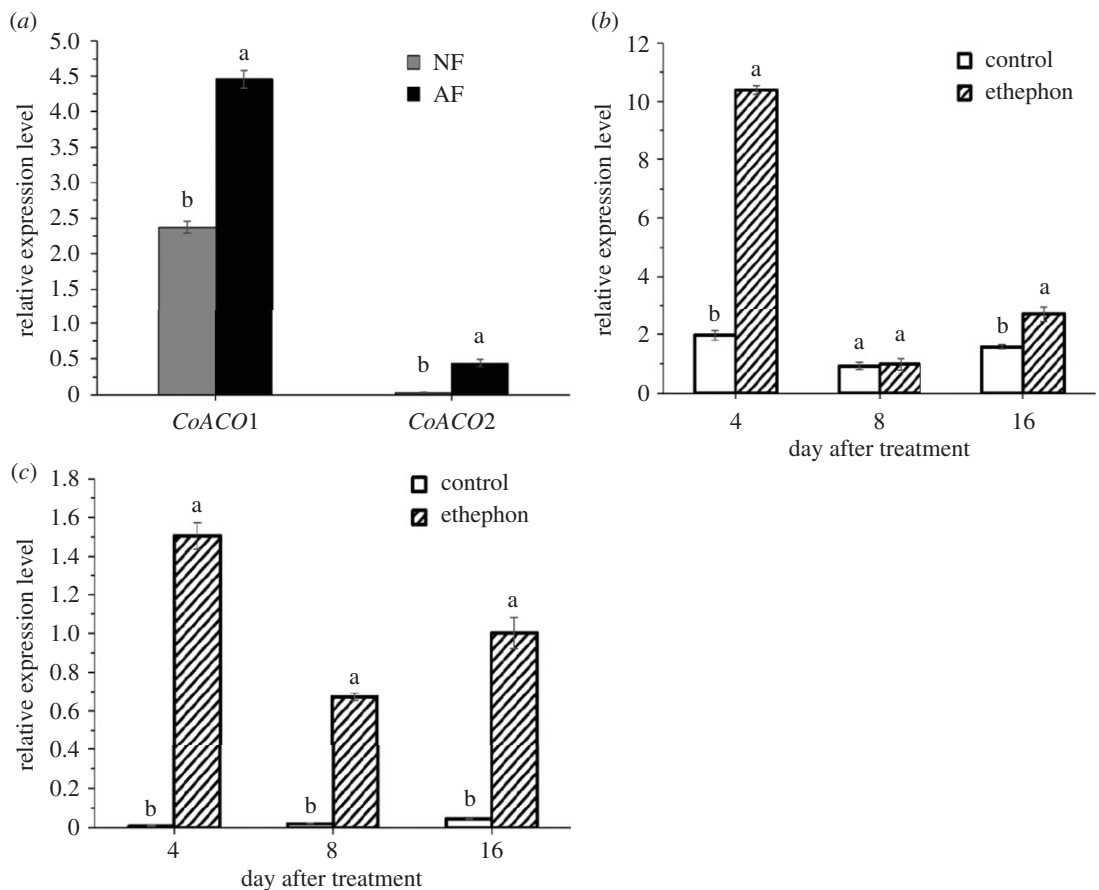

**Figure 6.** Relative expression levels of *CoACO* genes in FAZs of *C. oleifera* 'Huashuo' by qRT-PCR. (*a*) qRT-PCR analysis of *CoACO1* and *CoACO2* genes, respectively, in AZs of NF and AF. (*b,c*) qRT-PCR analysis of *CoACO1* and *CoACO2* genes in FAZs after the treatment of ethephon with respect to the control. Three biological replicates for each sample were averaged. Error bars indicate the s.d. of the means ($n = 3$). One-way ANOVA (Duncan's multiple range test) analysis was performed, and statistically significant differences ($p < 0.05$) were indicated by different letters.

*CoACO1* and *CoACO2* were relatively highly expressed in young leaves, fruits and FAZs (figure 5). These results suggest that *CoACO1* may be related to fruit ripening and *CoACO2* may be related to fruit abscission.

ACO is essential for ethylene synthesis in plants. To determine the relationship among *CoACO* genes, ethylene and fruit abscission in *C. oleifera*, the expression levels of *CoACO1* and *CoACO2* in AZs tissues of NF and AF during the August stage were conducted using a qRT-PCR assay. The results showed that the expression level of *CoACO1* and *CoACO2* in AZs of AF were significantly increased to about 1.9-fold and 14.7-fold higher than that in NF, respectively (figure 6*a*). Moreover, the expression level of *CoACO1* in AZs

tissues was significantly induced after ethephon treatment to about 5.3-fold and 1.7-fold higher than that in control on the 4th and 16th DAT, respectively (figure 6b). The expression level of CoACO2 in AZs of ETH-F was significantly increased to about 188.1-fold, 30.7-fold and 21.4-fold higher than that in control on the 4th, 8th and 16th DAT, respectively (figure 6c). These results indicate that CoACO1 and CoACO2 genes have an association with fruit abscission at the August stage in C. oleifera.

# 4. Discussion

## 4.1. Ethephon effectively stimulates fruit abscission in C. oleifera

Previous studies have reported that fruit abscission involves the ethylene synthesis in numerous plants [28–30]. The role of ethylene in fruit abscission was further corroborated using ethephon treatment [15,31]. In our study, the application of ethephon changed the surface structure of abscission fracture plane and promoted the fruit abscission in C. oleifera (figure 2), suggesting that the adhesion between fruit stalk and branch of ETH-F decreased compared to the NF. It is worth noting that the cumulative abscission rate in AZ of control remarkably increased by 17.5% from the 8th DAT to the 16th DAT (figure 1). We concluded that ethylene is a main cause of fruit abscission of the August stage in C. oleifera. With the development of C. oleifera industry, how to finish picking the fruit within a short time has been a long-standing problem that troubles people. In our further research, the ethephon can be used as a substance accelerating fruit shedding for the concentrated harvest of C. oleifera fruits at the same maturity stage.

The effect of ethephon on fruit abscission was different in different species. The fruit abscission rate was only 57.7% after the application of $2 \, \text{g} \, \text{l}^{-1}$ ethephon on the 16th DAT in C. oleifera 'Huashuo' (figure 1). However, 95% of fruitlets absorbed within 8 days after the ethephon treatment 7200 ppm in mango [32]. Thus, we reasonably speculated that fruit abscission in C. oleifera is a concentration-dependent response.

Moreover, effects of ethephon applications on plant drought tolerance have been reported. Foliar spraying with $400 \, \text{mg} \, \text{l}^{-1}$ ethephon could improve the drought tolerance of maize and accelerate the rehydration process after drought stress. The $100 \, \text{mg} \, \text{l}^{-1}$ ethephon-treated sugarcane showed stronger drought resistance than the controls [33]. In our study, from the 8th to 16th DAT, the relative fruit abscission rate of the ethephon was lower than that of the control (figure 1). It can be concluded that ethephon treatment enhanced drought tolerance of tea-oil trees and consequently reduced the fruit abscission caused by drought.

## 4.2. Ethylene plays an important role in fruit abscission in C. oleifera

ACC is transformed to ethylene under the catalysis of ACC oxidase. In our study, ACC content (figure 3a), expression level of CoACO1 and CoACO2 (figure 6a) and ACO activity (figure 3c) significantly increased in AZs of AF compared with that of NF which indicated that the whole ethylene biosynthesis pathway has been upregulated when fruits are close to abscission. In Lupinus luteus, the similar results were observed. The expression level of LlACO and ACC content were over two times higher in the pedicels of aborted flowers than the pedicels of normal flowers [30]. The effect of ethylene on the AZs structure was further confirmed by SEM. The AF will soon fall off. Thus, ethylene plays an important role in fruit abscission of AF in C. oleifera.

Due to the key role of ACO, encoded by ACO genes, in ethylene biosynthesis, controlling the expression of ACO genes could effectively regulate the ethylene biosynthesis. In our study, the treatment of ethephon showed a significant increase in ACO activity and ACC content (the 4th and 16th DAT) in FAZs (figure 3b,d). In addition, the CoACO genes expressions in the AZs of ETH-F were higher than that of the control (figure 6b,c). And the expression pattern of CoACO1 and CoACO2 genes showed good correspondence with fruit abscission under ethephon treatment. Zhu et al. [34] treated the fruitlets of 'Delicious' apple with NAA, and reported that the fruitlets abscission rate increased, and found that the expression of MdACO1 significantly increased in fruits and AZ. Taking our results together, we propose that ethylene is closely related to fruit abscission in C. oleifera.

## 4.3. CoACO genes cloning in C. oleifera

ACO genes, initially obtained from tomato [35], were gradually cloned from persimmon [36], poplar [37], pear [38] and peach [39]. ACO genes are members of multigene families. In model plant Arabidopsis

*thaliana*, 13 *ACO* genes were identified, among which *ACO1*, *ACO2* and *ACO4* were considered as functional *ACOs* in *Arabidopsis* [40]. Yuan *et al.* identified 11 *ACO* genes by exploring the pear genome. Among these genes, *ACO2*, *ACO3*, *ACO9* and *ACO11* are involved in pear fruit ripening [41]. But no related reports were found in *C. oleifera*.

In this study, we cloned the cDNAs of *CoACO1* and *CoACO2* for the first time. Two amino acid sequences of CoACO proteins contain the PLN02299 super family which is the conserved regions of *ACO* genes in other species (figure 4*b*). Multiple alignments of predicted CoACO proteins indicated that *CoACO1* has high homology with *VvACO1* and *CoACO2* has high homology with *VvACO2* (figure 4*a*,*c*). Similar structures may have similar physiological functions. *VvACO1* and *VvACO2* may have a role in the grape berry ripening phase [42]. In tomato, at least 7 *ACO* genes were identified, and *LeACO1* is the key gene related to ripening and leaf senescence [19,43]. This finding suggested *CoACO1* and *CoACO2* may have a role in fruit ripening and abscission.

# 5. Conclusion

In this study, the internal structure of the AZs of AF changed with respect to the NF and then affected the adhesion between the fruit stalk and stem. The ACC content and ACO activity were higher in AZs of AF than in that of NF. Exogenous ethephon treatment stimulated fruit abscission. The internal structure of AZs of ETH-F changed accordingly. The ACC content and ACO activity in AZs of ETH-F increased compared to that of NF. In general, ethylene plays an important role in fruit abscission at the August stage in *C. oleifera*. In addition, two *ACO* genes, namely *CoACO1* and *CoACO2*, were isolated from *C. oleifera* for the first time. The qRT-PCR analysis confirmed that the expression level of *CoACO1* and *CoACO2* upregulated significantly in AZs of AF and ETH-F on the 4th and 16th DAT. Our results suggested that *CoACO* genes are involved in the regulation of fruit abscission at the August stage in *C. oleifera* by ethylene.

Data accessibility. Data available from https://doi.org/10.6084/m9.figshare.14385731. The ACO protein sequences used for phylogenetic analysis are given in electronic supplementary material, S1. Raw data of figures 1, 3, 5 and 6 are given in electronic supplementary material, S2.

Authors' contributions. D.Y. and X.M. conceived and designed the experiments. X.H. and H.L. performed the experiments. M.Y. and J.Z. analysed the data. X.H., M.Y. and X.M. wrote the paper. M.S. and S.G. revised the paper. All authors have read and approved the manuscript.

Competing interests. The authors declare that the research was conducted in the absence of any commercial or financial relationships that could be construed as a potential conflict of interest.

Funding. This work was supported by The National Key R&D Program of China (2018YFD1000603) and Scientific Research Foundation for Advanced Talents of Central South University of Forestry and Technology (2019YJ003).

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
