## [Peer Review File · Royal Society Open Science]

Review History

RSOS-202340.R0 (Original submission)

Review form: Reviewer 1

Is the manuscript scientifically sound in its present form?

Yes

Are the interpretations and conclusions justified by the results?

Yes

Is the language acceptable?

Yes

Do you have any ethical concerns with this paper?

No

Have you any concerns about statistical analyses in this paper?

No

Recommendation?

Accept with minor revision (please list in comments)

Comments to the Author(s)

The following are minor corrections.

1. Please delete sheets 2 and 3 in the supplementary file 3 as sheet 1 includes all the information needed;
2. Please submit the CDS sequence of CoACO1 and CoACO2 to GenBank and provide the GenBank ID in the manuscript instead of putting the sequences in supplementary file 1;
3. In Figure 2, please highlight the Arabic alphabets and arrows with colors.
4. In Figure 3, please add 'A' at the first column diagram;
5. The GeneBank numbers showed in Fig. 4C are not the same as the ones in "Supplementary file 2". Please double check.
6. Numbers should be in academic writing, for example, 1057 bp in Table 1 should be 1,057 bp;
7. In Line 124, could you explain why the optimal ethephon concentration is 2 g/L? References might be needed here to show why this concentration is used;
8. Line 208, 'On 4th DAT' should be revised as "On the 4th DAT";
9. In line 264, please explain a little bit why you choose ACO1, ACO2, and ACO3 from grape to the protein alignment with CoACO1 and CoACO2. Similarly, please also explain the reason for using full-length ACO protein sequences from Arabidopsis, grape, tomato, sorghum, and rice in the phylogenetic analysis;
10. Lines 595 and 604, 'Relative levels expression' and 'Relative levels of expression' should be revised as 'Relative expression levels'.

Review form: Reviewer 2

Is the manuscript scientifically sound in its present form?

Yes

Are the interpretations and conclusions justified by the results?

Yes

Is the language acceptable?

No

Do you have any ethical concerns with this paper?

No

Have you any concerns about statistical analyses in this paper?

No

Recommendation?

Reject

Comments to the Author(s)

In this manuscript, the authors investigate the role of ethylene in fruit abscission in *Camellia olifera*. This is an important subject, but I feel like the role of ethylene in this response is already well characterised - as the authors discuss in their introduction.

Major points:

1) My main concern with the manuscript is that is rather lacking in novelty; the role of ethylene in fruit abscission is well described, and although the authors describe it in *Camellia* for the first time here, none of this is a surprise. If the authors added some new insight into the role of ethylene in fruit abscission, it would make this a much better manuscript.

2) Ethephon

The authors use ethephon, but never tell the reader what ethephon is, or does.

3) Sample size, statistics

From the authors descriptions, it is impossible to tell whether -

a) each experiment was performed with three samples, and all data are presented in supplementary file 3

or

b) each experiment was performed 3 times, and only the experimental means of each experiment are reported in supplementary file 3.

This is a very big distinction, and needs clarifying. If each experiment was performed 3 times, what was the sample size of each experiment?

All data were analysed using ANOVA, but it seems unlikely that with $n=3$, that all data were normally distributed. Did the authors check this before performing ANOVA?

Decision letter (RSOS-202340.R0)

Dear Mrs Ma

On behalf of the Editors, we are pleased to inform you that your Manuscript RSOS-202340 "Ethylene effects immature fruits abscission is associated with higher expression of CoACO genes in *Camellia oleifera*" has been accepted for publication in Royal Society Open Science subject to minor revision in accordance with the referees' reports. Please find the referees' comments along with any feedback from the Editors below my signature.

One reviewer is very positive, but each reviewer raises a number of points that will require careful attention and addressing in a revised version. We invite you to respond to the comments and revise your manuscript. Below the referees' and Editors' comments (where applicable) we provide additional requirements. Final acceptance of your manuscript is dependent on these requirements being met. We provide guidance below to help you prepare your revision.

Please submit your revised manuscript and required files (see below) no later than 7 days from today's (ie 31-Mar-2021) date. Note: the ScholarOne system will 'lock' if submission of the revision is attempted 7 or more days after the deadline. If you do not think you will be able to meet this deadline please contact the editorial office immediately.

Please note article processing charges apply to papers accepted for publication in Royal Society Open Science (<https://royalsocietypublishing.org/rsos/charges>). Charges will also apply to papers transferred to the journal from other Royal Society Publishing journals, as well as papers submitted as part of our collaboration with the Royal Society of Chemistry

(<https://royalsocietypublishing.org/rsos/chemistry>). Fee waivers are available but must be requested when you submit your revision (<https://royalsocietypublishing.org/rsos/waivers>).

on behalf of Dr James Locke (Associate Editor) and Steve Brown (Subject Editor)
openscience@royalsociety.org

Associate Editor Comments to Author (Dr James Locke):

The manuscript has been examined by 2 expert reviewers. The reviewers point out a series of corrections that need to be fixed in a revised manuscript. The concern of one reviewer over the level of novelty in the work do not need to be addressed.

Please attempt to improve the grammar in the manuscript as requested by reviewer 2. In particular the grammar in the title 'Ethylene effects immature fruits abscission is associated with higher expression of CoACO genes in *Camellia oleifer*' should be checked.

Reviewer comments to Author:

Reviewer: 1

Comments to the Author(s)

The following are minor corrections.

1. Please delete sheets 2 and 3 in the supplementary file 3 as sheet 1 includes all the information needed;
2. Please submit the CDS sequence of CoACO1 and CoACO2 to GenBank and provide the GenBank ID in the manuscript instead of putting the sequences in supplementary file 1;
3. In Figure 2, please highlight the Arabic alphabets and arrows with colors.
4. In Figure 3, please add 'A' at the first column diagram;
5. The GeneBank numbers showed in Fig. 4C are not the same as the ones in "Supplementary file 2". Please double check.
6. Numbers should be in academic writing, for example, 1057 bp in Table 1 should be 1,057 bp;
7. In Line 124, could you explain why the optimal ethephon concentration is 2 g/L? References might be needed here to show why this concentration is used;
8. Line 208, 'On 4th DAT' should be revised as "On the 4th DAT";
9. In line 264, please explain a little bit why you choose ACO1, ACO2, and ACO3 from grape to the protein alignment with CoACO1 and CoACO2. Similarly, please also explain the reason for using full-length ACO protein sequences from Arabidopsis, grape, tomato, sorghum, and rice in the phylogenetic analysis;
10. Lines 595 and 604, 'Relative levels expression' and 'Relative levels of expression' should be revised as 'Relative expression levels'.

Reviewer: 2

Comments to the Author(s)

In this manuscript, the authors investigate the role of ethylene in fruit abscission in *Camellia olifera*. This is an important subject, but I feel like the role of ethylene in this response is already well characterised - as the authors discuss in their introduction.

Major points:

1) My main concern with the manuscript is that it is rather lacking in novelty; the role of ethylene in fruit abscission is well described, and although the authors describe it in *Camellia* for the first time here, none of this is a surprise. If the authors added some new insight into the role of ethylene in fruit abscission, it would make this a much better manuscript.

2) Ethephon

The authors use ethephon, but never tell the reader what ethephon is, or does.

3) Sample size, statistics

From the authors descriptions, it is impossible to tell whether -

a) each experiment was performed with three samples, and all data are presented in supplementary file 3

or

b) each experiment was performed 3 times, and only the experimental means of each experiment are reported in supplementary file 3.

This is a very big distinction, and needs clarifying. If each experiment was performed 3 times, what was the sample size of each experiment?

All data were analysed using ANOVA, but it seems unlikely that with $n=3$, that all data were normally distributed. Did the authors check this before performing ANOVA?

===PREPARING YOUR MANUSCRIPT===

===PREPARING YOUR REVISION IN SCHOLARONE===

Author's Response to Decision Letter for (RSOS-202340.R0)

See Appendices A & B.

Decision letter (RSOS-202340.R1)

Dear Mrs Ma,

It is a pleasure to accept your manuscript entitled "Ethylene-regulated immature fruit abscission is associated with higher expression of CoACO genes in *Camellia oleifera*" in its current form for publication in Royal Society Open Science. The comments of the reviewer(s) who reviewed your manuscript are included at the foot of this letter.

Kind regards,
Royal Society Open Science Editorial Office
Royal Society Open Science

on behalf of Dr James Locke (Associate Editor) and Steve Brown (Subject Editor)
openscience@royalsociety.org

Associate Editor Comments to Author (Dr James Locke):

Comments to the Author:

The revision addresses the issues raised by the reviewers and is suitable for publication.

Appendix A

Key Laboratory of Non-Wood Forest Products of Forestry Ministry, Central South University
of Forestry and Technology, Changsha, Hunan, 410004, China

April 5, 2021

Dear Editor,

Many thanks for you editing our manuscript.

We received your email about our submitted manuscript RSOS-202340 “Ethylene effects immature fruits abscission is associated with higher expression of *CoACO* genes in *Camellia oleifera*” on Mar. 31, 2021, which needs to be revised before reconsidered acceptance for publication in Royal Society Open Science. We sincerely appreciate all your insightful suggestions and comments. All the questions and comments have been carefully considered and replied. We are sorry for our language mistakes. The software “<https://www.crimsoni.ai/editcheck/>” was used to correct the language errors in the revised manuscript. In addition, the author Dr. Muhammad Sajjad whose official language is English has corrected some language errors in the revised manuscript. The title ‘Ethylene effects immature fruits abscission is associated with higher expression of *CoACO* genes in *Camellia oleifer*’ has been revised as ‘Ethylene-regulated immature fruit abscission is associated with higher expression of *CoACO* genes in *Camellia oleifera*’. We fervently hope that the revised manuscript is now acceptable for publication.

If you have any further questions or comments, please let me know.

With my best regards!

Yours sincerely

Xiaoling Ma (Ph D)

On behalf of all the authors

Key Laboratory of Cultivation and Protection for Non-Wood Forest Trees of Ministry of Education,

Key Laboratory of Non-Wood Forest Products of Forestry Ministry, Central South University of
Forestry and Technology,

Changsha, Hunan, 410004, China,

E-mail: fanxiaoling@163.com

Appendix B

Dear Editor,

Many thanks for reviewing our manuscript. All points you raised have been considered carefully and answered. The title “Ethylene effects immature fruits abscission is associated with higher expression of *CoACO* genes in *Camellia oleifera*” has been revised as “Ethylene-regulated immature fruit abscission is associated with higher expression of *CoACO* genes in *Camellia oleifera*”. All line numbers of the revision referenced in the manuscript are from the revised manuscript (the highlighted copy).

Reviewer: 1

1. Please delete sheets 2 and 3 in the supplementary file 3 as sheet 1 includes all the information needed.

Answer: Thanks for your suggestion. We have deleted sheets 2 and 3 in the revised supplementary file 3 which has been renamed supplementary file 2.

2. Please submit the CDS sequence of CoACO1 and CoACO2 to GenBank and provide the GenBank ID in the revised manuscript instead of putting the sequences in supplementary file 1.

Answer: Thanks for your suggestion. We have submitted the CDS sequences of CoACO1 and CoACO2 to GenBank and provided the GenBank ID (CoACO1 (MW854295), CoACO2 (MW854296)) in the revised manuscript (L.618) and Fig. 4C. Meantime, the original supplementary file 1 have been deleted in the revised manuscript (L.279, L.569, L.617).

3. In Figure 2, please highlight the Arabic alphabets and arrows with colors.

Answer: Thanks for your suggestion. We have highlighted the Arabic alphabets and arrows with colors in the revised manuscript (Fig. 2).

4. In Figure 3, please add ‘A’ at the first column diagram;

Answer: We are sorry for our mistake. We have added ‘A’ at the first column diagram in Fig. 3 in the revised manuscript.

5. The GeneBank numbers showed in Fig. 4C are not the same as the ones in “Supplementary file 2”. Please double check.

Answer: We are sorry for our mistake. We have changed the GenBank numbers showed in “Supplementary file 2” to the same as those in Fig. 4C in the revised manuscript. Due to the deletion of “Supplementary file 1”. In this part, the “Supplementary file 2” has been renamed as “Supplementary file 1”.

6. Numbers should be in academic writing, for example, 1057 bp in Table 1 should be 1,057 bp;

Answer: We are sorry for our mistake. We have corrected the correspond numbers in academic writing in the revised manuscript (L.578-579, L.203-204).

7. In Line 124, could you explain why the optimal ethephon concentration is 2 g/L? References might be needed here to show why this concentration is used;

Answer: We are sorry for our incomplete statement. In the previous experiment, we used different concentrations of ethephon to spray *C. oleifera*. The results showed that when using less than 2 g L⁻¹ ethephon, *C. oleifera* could not show obvious fruit abscission, and when using more than 2 g L⁻¹ ethephon, the tea oil trees would be damaged. However, when using 2 g L⁻¹ ethephon, the fruit abscission rate of *C. oleifera* was significantly increased and the trees were not damaged, so we sprayed the *C. oleifera* with 2 g L⁻¹ ethephon.

In the revised manuscript, the statement “Three ‘Huashuo’ trees were randomly selected for each of the following treatments: water (control) and 2 g L⁻¹ ethephon solution. ” has been changed to “Three ‘Huashuo’ trees were randomly selected for each of the following treatments: water (control) and 2 g L⁻¹ ethephon solution (Earlier experimental results showed that 2 g L⁻¹ ethephon could significantly increase the fruit abscission rates as compared to the control and the trees were not damaged).” (L.136-139)

8. Line 208, ‘On 4th DAT’ should be revised as “On the 4th DAT”.

Answer: Thank you for your suggestion. Line 208, the description of “On the 4th DAT” is right in our manuscript (L.225).

9. In line 264, please explain a little bit why you choose ACO1, ACO2, and ACO3 from grape to the protein alignment with CoACO1 and CoACO2. Similarly, please also explain the reason for using full-length ACO protein sequences from Arabidopsis,

grape, tomato, sorghum, and rice in the phylogenetic analysis.

Answer: Thank you for this comment. The coding sequences of *CoACO* genes were used as a query for BLAST searchers to acquire the *ACO* genes that have been reported in other species. It was found that the *ACO* genes from Arabidopsis, grape, tomato, sorghum, and rice had a certain similarity with *CoACO* genes and therefore we used these *ACO* proteins for the phylogenetic analysis. The phylogenetic analysis showed that the amino acid sequence similarity between *ACOs* from grape and *CoACOs* were relatively high, so we chose *ACO1*, *ACO2*, and *ACO3* from grape to the protein alignment with *CoACO1* and *CoACO2*.

10. Lines 595 and 604, 'Relative levels expression' and 'Relative levels of expression' should be revised as 'Relative expression levels'.

Answer: We are sorry for our mistakes. "Relative levels expression" and "Relative levels of expression" have been revised as "Relative expression levels" in the revised manuscript (L.625, L.634).

Reviewer: 2

Point 1: My main concern with the manuscript is that is rather lacking in novelty; the role of ethylene in fruit abscission is well described, and although the authors describe it in *Camellia* for the first time here, none of this is a surprise. If the authors added some new insight into the role of ethylene in fruit abscission, it would make this a much better manuscript.

Answer: Thanks for your suggestion. Ethylene and abscisic acid are widely recognized as hormones that promote the abscission of plant organs. However, our previous experiments showed that there was no significant difference in abscisic acid content between abscission zone of normal fruit and that of abnormal fruit, while ethylene did. So we explored the important role of ethylene in the fruit abscission of *Camellia oleifera* in this study, and we are going to conduct an in-depth study on how ethylene regulates the fruit abscission of *C. oleifera* in the next stage.

Point 2: Ethephon.

The authors use ethephon, but never tell the reader what ethephon is, or does.

Answer: We are sorry for our incomplete statement. “Ethephon (2-chloroethylphosphate acid) is an ethylene-releasing molecule which is stable in a low pH solution, but it hydrolyses in the higher pH of plant tissues releasing ethylene (Ferrara et al., 2016). Ethephon is a gaseous plant growth regulator that can promote fruit ripening, affect fruit color, firmness, carbon dioxide production and other physiological properties (Zhang et al., 2012). Exogenously applied ethephon stimulates ethylene production and triggers ethylene-dependent reactions such as flower or fruit abscission (Wertheim, 2000)”, this description and correspond references have been added to the revised manuscript (L.75-82).

Point 3: Sample size, statistics

From the authors descriptions, it is impossible to tell whether -

- a) each experiment was performed with three samples, and all data are presented in supplementary file 3 or
- b) each experiment was performed 3 times, and only the experimental means of each experiment are reported in supplementary file 3.

This is a very big distinction, and needs clarifying. If each experiment was performed 3 times, what was the sample size of each experiment? All data were analysed using ANOVA, but it seems unlikely that with $n=3$, that all data were normally distributed. Did the authors check this before performing ANOVA.

Answer: We are sorry for our incomplete statement. Each experiment was performed with three samples and each sample has three technical replicates in our study. And all data presented in supplementary file 3 were obtained by taking the average of three biological and technical replicates. We have checked the data before performing the ANOVA.